# Evidence to Date: Evaluating Pembrolizumab in the Treatment of Extensive-Stage Small-Cell Lung Cancer

**Ivy Riano [1], Shruti R. Patel [2], Stephen V. Liu [3] and Narjust Duma [4],***

[1] Department of Medicine, MetroWest Medical Center, Tufts University School of Medicine, Framingham, MA 01702, USA; ivy.riano@dartmouth.edu
[2] Department of Medicine, Mayo Clinic, Rochester, MN 55902, USA; Patel.Shruti@mayo.edu
[3] Department of Medical Oncology, Lombardi Comprehensive Cancer Center, Georgetown University, Washington, DC 20057, USA; Stephen.Liu@gunet.georgetown.edu
[4] Division of Medical Oncology, Hematology and Palliative Care, University of Wisconsin, Madison, WI 53706, USA
* Correspondence: nduma@wisc.edu; Tel.: +1-608-2653837

**Abstract:** Small-cell lung cancer (SCLC) is an aggressive subtype of lung cancer characterized by a rapid initial response and early development of resistance to systemic therapy and radiation. The management of SCLC significantly changed for the first time in decades with the introduction of immune checkpoint inhibitors. Pembrolizumab, a humanized IgG4 isotype antibody, targets the programmed cell death protein 1 (PD-1) pathway to restore anti-tumor immunity. Prospective trials of pembrolizumab in patients with previously treated SCLC showed significant durability of responses. These results led to the U.S. Food and Drug Administration (FDA) granting pembrolizumab accelerated approval as second- or third-line monotherapy for patients with extensive-stage (ES) SCLC. In a recent clinical trial that included patients with previously untreated ES-SCLC, pembrolizumab in combination with platinum/etoposide met its progression-free survival endpoint, but overall survival (OS) did not cross the threshold for superiority. With the therapeutic landscape for SCLC rapidly evolving, we review prior experience and future directions of pembrolizumab in ES-SCLC.

**Keywords:** small-cell lung cancer; pembrolizumab; immunotherapy; PD-1; checkpoint inhibitor

## 1. Introduction

Small-cell lung cancer (SCLC) is a neuroendocrine tumor that represents about 13% of all lung cancers and occurs predominantly in smokers [1]. In general, SCLC grows rapidly and has high metastatic potential. These two properties contribute to a particularly high mortality rate. Most patients have advanced disease and present with distant metastases, malignant effusions, and/or contralateral supraclavicular or hilar lymph node involvement. In these patients, systemic chemotherapy is typically the primary therapeutic modality, with some patients also drawing benefit from radiation therapy [2]. Although the tumor–node–metastasis (TNM) classification is preferred to the staging system of the Veterans Administration Lung Study Group (VALSG), which separates limited-stage (LS) disease (tumor confined to one hemi-thorax and one radiation port; no malignant pleural or pericardial effusion) from extensive-stage (ES) disease (not meeting criteria for LS), the latest staging system is still widely used in both designing clinical trials and presenting data from them, as it effectively distinguishes patients treated primarily with chemotherapy (LS disease) from those treated with systemic chemotherapy or chemoimmunotherapy (ES disease) [3–5].

The initial approach to SCLC treatment varies substantially by stage. In non-metastatic SCLC, the therapeutic goals are to achieve durable control of thoracic disease and reduce the risk of metastatic dissemination. Local treatment options include surgery and radiotherapy. Chemotherapy can both augment the local efficacy of radiation and potentially treat

micrometastatic disease. The standard chemotherapy regimen in this setting is cisplatin–etoposide. In patients who respond to initial treatment, prophylactic cranial irradiation (PCI) is also part of the standard management with non-metastatic disease [3,6]. For ES disease, the first-line chemotherapy for newly diagnosed metastatic SCLC consisted of a platinum agent (cisplatin or carboplatin) with etoposide. Radiotherapy is traditionally reserved for the palliation of symptoms in patients with ES disease, and PCI remains controversial [3,7]. Despite a typically dramatic initial response to therapy, most patients with SCLC experience relapse within 6 months despite chemotherapy and radiation and the 5-year survival rates for patients with ES disease remain low (5%) [2]. As outcomes are poor, therapeutic options after relapse are limited.

Several groups have pursued comprehensive genomic profiling of SCLC with hopes of identifying actionable genomic targets. Studies have shown high somatic mutation rates and copy number alterations in the tumor tissues with a near-universal bi-allelic inactivation of TP53 and RB1 [8,9]. Unfortunately, there has been a notable lack of activating mutations in driver oncogenes in SCLC, and molecularly targeted agents have yet to find a place in the treatment of SCLC. This is not to say alterations are uncommon; a comprehensive analysis of 236 cancer genes in 98 patients using next-generation sequencing demonstrated that all patients had at least one genomic alteration, and an average of 3.9 alterations was seen per tumor [10]. These high mutation rates suggest that these tumors may respond to immune checkpoint inhibition, as previous work has correlated the rate of somatic mutations to the efficacy with programmed cell death protein 1 (PD-1) inhibitors [11].

The advent of immune checkpoint inhibitors has changed the landscape of oncology care over the past decade. The mainstay of treatment for ES-SCLC has been platinum-based chemotherapy with etoposide [12]. Nivolumab monotherapy was the first immune checkpoint inhibitor to show durable responses, leading to its accelerated approval in the United States (U.S.) as third-line monotherapy in patients with ES-SCLC [13]. Pembrolizumab showed similar results, leading to its accelerated approval as third-line monotherapy [14].

The interest in immunotherapy for SCLC, however, is undeniable. This is largely based on the durability of response and the potential for long-term survival. These features coupled with the high attrition rate seen in SCLC [15] prompted the earlier introduction of immunotherapy in subsequent prospective trials. Two randomized phase 3 clinical trials demonstrated an improvement in survival with the addition of an anti-programmed death-ligand 1 (PD-L1) antibody, either atezolizumab or durvalumab, to standard first-line platinum-based chemotherapy [16,17]. These outcomes led to the U.S. Food and Drug Administration (FDA) full approval of atezolizumab and durvalumab in the first-line setting. Pembrolizumab given with platinum-doublet chemotherapy was also studied in the first-line setting but did not demonstrate a survival benefit [18]. In this article, we review the evidence supporting the use of pembrolizumab in the management of patients with ES-SCLC and describe how past experience can guide future use.

## 2. Pembrolizumab for Previously-Treated ES-SCLC

Pembrolizumab is a selective, humanized, monoclonal anti-PD-1 antibody that disrupts the interaction between the PD-1 and its ligand, PD-L1, allowing activation and expansion of cytotoxic T-cells to facilitate an immune-mediated, anti-tumor response [17]. Pembrolizumab has demonstrated relevant clinical activity in patients with previously treated ES-SCLC. Most of the evidence regarding the use of pembrolizumab in previously treated SCLC comes from two single-arm trials—KEYNOTE-028 cohort C1 (NCT02054806) [19] and KEYNOTE-158 cohort G1 (NCT02628067) [20]. A comparison between the designs of pembrolizumab clinical trials in ES-SCLC is included in Table 1.

The phase 1b open-label KEYNOTE-028 was a multicohort trial that explored the safety and efficacy of pembrolizumab in patients with various PD-L1-positive tumors [19]. In this study, patients received pembrolizumab 10 mg/kg every two weeks for 24 months or until documented disease progression or intolerable toxicity. Cohort C1 included patients

with ES-SCLC or primary pulmonary neuroendocrine tumors that had failed standard therapy. Tumor PD-L1 expression assayed by immunohistochemistry (IHC) using the Dako 22C3 PD-L1 clone was required for entry. PD-L1 positivity was defined by membranous PD-L1 expression in ≥1% of tumor cells and associated inflammatory cells or positive staining in stroma [19].

**Table 1.** Comparison between designs of published pembrolizumab clinical trials in extensive-stage small-cell lung cancer.

| Trial | Design | End Points | PD-L1 Expression | Key Eligibility Criteria | Pembrolizumab Dose | Response Assessment |
|-------|--------|-----------|------------------|--------------------------|--------------------|--------------------| 
| KEYNOTE-028 [19] | Multicohort Phase 1b open-label for previously treated SCLC | Primary: ORR; secondary: PFS, OS, DOR, safety, and tolerability | PD-L1 expression was required | SCLC or pulmonary neuroendocrine tumor that had failed standard therapy | Pembrolizumab 10 mg/kg every 2 weeks | Every 8 weeks for 6 months; every 12 weeks thereafter |
| KEYNOTE-158 [20] | Multicohort Phase 2 open-label for previously treated SCLC | Primary: ORR; secondary: PFS, OS, DOR, and safety | No PD-L1 expression required | Evaluable tumor sample for biomarker assessments | Pembrolizumab 200 mg IV every 3 weeks | Every 9 weeks for 12 months; every 12 weeks thereafter |
| Gadgeel et al. [21] | Phase 2 open-label, single-arm maintenance pembrolizumab after 1st line chemotherapy | Primary: PFS; secondary: OS and safety | No PD-L1 expression required | Response or stable disease after chemotherapy and enrollment within 8 weeks of last chemotherapy dose | Pembrolizumab 200 mg IV every 3 weeks | Every 6 weeks (two cycles) for the first six cycles and then at the discretion of the treating physician |
| KEYNOTE-604 [18] | Phase 3 randomized, double-blind, placebo-controlled for the 1st-line treatment of ES-SCLC | Primary: PFS, OS; secondary: ORR, DOR, and safety | PD-L1 expression was assessed retrospectively | SCLC not previously treated with systemic therapy | Pembrolizumab 200 mg IV every 3 weeks + platinum/etoposide | At baseline, every 6 weeks for the first 48 weeks, and every 9 weeks thereafter |
| Kim et al. [22] | Phase 2, multi-center, open label, single-arm for ES-SCLC that had not responded to 1st line | Primary: ORR; secondary: OS, PFS, safety and analysis of biomarkers | PD-L1 expression was required | ED SCLC that progressed after 1st line standard treatment regardless of their initial best response | Pembrolizumab 200 mg IV every 3 weeks + paclitaxel | At baseline, every two cycles until six cycles. Thereafter, every three cycles |

ORR, objective response rate; PFS, progression-free survival; OS, overall survival; DOR, duration of response; PD-L1, programmed death-ligand 1; SCLC, small-cell lung cancer; mg, milligrams; kg, kilograms; IV, intravenous.

A total of 163 patients were screened for enrollment, 31.7% tested positive for PD-L1 expression, and 24 patients were treated. At the time the data was presented, the study median follow-up was 9.8 months (range, 0.5–24.4 months). Patients were heavily pretreated; 87.5% of patients had received two or more lines of therapy. The confirmed overall response rate (ORR) was 33.3% (95% confidence interval (CI), 15.6–55.3), and the median duration of response (DOR) was 19.4 months (range, ≥3.6–≥20.0 months) with three patients remaining in treatment at the time of data cutoff. The median progression-free survival (PFS) was limited to 1.9 months (95% CI, 1.7–5.9), but the median OS was 9.7 months (95% CI, 4.1—not reached). Treatment-related adverse events (AEs) were seen in 16 (66.7%) of 24 patients. Eight patients (33.3%) had grade 3 to 5 AEs, two of whom had AEs related to treatment. One patient experienced grade 3 bilirubin elevation, and another patient experienced grade 3 asthenia and grade 5 colitis/intestinal ischemia [19].

This encouraging activity prompted development of the larger KEYNOTE-158 study—a phase 2 open-label multi-cohort study of eleven cancer types, including a cohort for patients with ES-SCLC who had progressed after or were ineligible for standard ther-

apy [20]. Having an evaluable tumor sample for biomarker assessment was an eligibility criterion, but tumor PD-L1 expression was not required. The pembrolizumab dose in this trial was a fixed dose of 200 mg every three weeks. A total of 107 patients with ES-SCLC were included in the study; 36 patients (34%) were continuing on-study at the data cutoff date. Again, patients were heavily pretreated; 79% had one or two prior therapies. Median follow-up was 10.1 months (range, 0.5–17.5). Tumors were PD-L1–positive in 42 patients (39%); this was determined using the combined positive score (CPS), defined as the ratio of PD-L1–positive cells (including tumor cells, lymphocytes, and macrophages) to the total number of tumor cells × 100. PD-L1 positivity was defined as a CPS ≥ 1. Of the total study population, 14% were unevaluable for CPS evaluation. The ORR with pembrolizumab in this trial was modest at 18.7% (95% CI, 11.8–27.4). Differences in response were observed in patients with PD-L1–positive tumors (using CPS) with an ORR of 35.7% (95% CI, 21.6–52.0) versus 6.0% (95% CI, 1.3–16.5) in patients with PD-L1-negative tumors. Median DOR had not been reached at the time of data cutoff (range, 2.1–18.7 months), but twelve patients had a DOR of over 9 months. Median OS was 8.7 months and median PFS was 2.0 months (95% CI, 1.9–2.1) in all patients, without significant difference in PFS between patients with PD-L1-positive and PD-L1-negative tumors (2.1 versus 1.9 months). Treatment-related AEs occurred in 63 patients (59%) and led to four treatment discontinuations and one death (pneumonia) [20].

Results from a pooled analysis of these two clinical trials, KEYNOTE-028 cohort C1 and KEYNOTE-158 cohort G1 are described in Table 2 [14]. Of the 131 patients included from both cohorts, this pooled efficacy and safety analysis included 83 patients with ES-SCLC (19 from KEYNOTE-028 and 64 from KEYNOTE-158) who had previously received ≥2 lines of therapy for advanced disease. Including both trials, 47 patients (57%) had PD-L1-positive tumors, and 30 (36%) had received ≥3 lines of therapy. In the third-line-and-beyond setting, the ORR to pembrolizumab was 19.3% (95% CI, 11.4–29.4). Two patients (2.4%) had a complete response, and 14 had a partial response; 14 of 16 responders (88%) had PD-L1-sitive tumors. Median DOR was not reached (range, 4.1–35.8 months). Median time to response was 2.1 (range, 1.7–4.1) months; 54% of patients had disease progression at the time of data cutoff. Median PFS was 2 months (95% CI, 1.9–3.4). The median OS with pembrolizumab was 7.7 months (95% CI, 5.2–10.1), with impressive landmark survival rates at 12 months (34.3%) and 24 months (20.7%). In the sum of AEs, no significant differences were observed between patients with one or two lines of prior therapy. Treatment-related AEs occurred in 83 patients (61.4%). The most common adverse events included fatigue (12%), pruritus (12%), rash (12%), hypothyroidism (10.8%), and arthralgia (9.6%). Grade 3 immune-related AEs (7.2%) included colitis, severe skin reaction, adrenal insufficiency, pneumonitis, and pancreatitis [14].

**Table 2.** Summary KEYNOTE-028 and KEYNOTE-158 trial results.

| Clinical Study | ORR | DOR | PFS | OS |
|---|---|---|---|---|
| KEYNOTE-028 [19] | 33.3% (95% CI, 15.6–55.3) | 19.4 mo (range, 3.6–20.0) | 1.9 mo (95% CI, 1.7–5.9) | 9.7 mo (range, 4.1-NR) |
| KEYNOTE-158 [20] | 18.7 % (95% CI, 11.8–27.4) | NR (range, 2.1–18.7) | 2.0 mo (95% CI, 1.9–2.1) | 8.7 mo |
| Pooled analysis * [14] | 19.3% (95% CI, 11.4–29.4) | NR (range, 4.1–35.8) | 2.0 mo (95% CI, 1.9–3.4) | 7.7 mo (95% CI, 5.2–10.1) |

* Pooled analysis: KEYNOTE-028 and KEYNOTE-158. ORR, objective response rate; PFS, progression-free survival; OS, overall survival; DOR, duration of response; CI, confidence interval; mo, months; NR, not reached.

In summary, the pooled analysis suggested pembrolizumab had antitumor activity among patients with ES-SCLC who had received ≥2 previous lines of therapy, regardless of PD-L1 expression. Responses were durable for 12 months or longer in 67.7% of patients, and 18 months or longer in 60.9% of the responders [14]. Based on the durability of the response, the U.S. FDA granted accelerated approval to pembrolizumab as third-line

treatment for patients with ES-SCLC and disease progression on or after platinum-based chemotherapy and at least one other prior line of therapy.

### 3. Pembrolizumab for Previously Untreated ES-SCLC

While the approval of pembrolizumab (and nivolumab) in the third-line setting was an important landmark for SCLC, the greatest impact for this disease has been the integration of immunotherapy in the first-line setting. Two randomized trials have shown that the addition of a PD-L1 inhibitor to first-line platinum plus etoposide improves survival. In the double-blind, placebo-controlled IMpower 133 trial (NCT02763579), the addition of atezolizumab to carboplatin plus etoposide improved both PFS and OS, leading to its approval by the U.S. FDA in March 2019 [16]. In the open-label CASPIAN study (NCT03043872), durvalumab added to platinum plus etoposide improved OS to a similar degree, leading to its approval in March 2020 [17]. KEYNOTE-604 (NCT03066778) explored the impact of adding pembrolizumab to platinum plus etoposide [18]. While the study design was fairly similar to the other two trials, the results, unfortunately, were not; KEYNOTE-604 failed to demonstrate a survival advantage (Table 3).

Similar to IMpower 133 and CASPIAN, KEYNOTE-604 was a global randomized trial for patients with treatment-naïve ES-SCLC. Patients with brain metastases were eligible if lesions were treated at least 14 days before study entry [18]. CASPIAN permitted untreated brain metastases [17] and IMpower 133 included only treated brain metastases [16] but unlike KEYNOTE-604, did not mandate time between treatment and study entry. Like CASPIAN, KEYNOTE-604 permitted either cisplatin or carboplatin to be paired with etoposide. Patients received four cycles of platinum plus etoposide and were randomized 1:1 to receive concurrent pembrolizumab at a fixed dose of 200 mg every 3 weeks or placebo for up to 35 cycles. Stratification factors were choice of platinum, ECOG performance status, and baseline lactate dehydrogenase concentration. The co-primary endpoints were PFS and OS [18].

A total of 453 patients were randomized; 223 of 228 participants were treated with at least $\geq 1$ dose of pembrolizumab plus chemotherapy and 222 of 225 with the placebo and chemotherapy. At a median follow-up of 21.6 months, 9% of patients in the pembrolizumab group and 1.4% in the placebo group remained on study treatment. The ORR was 71% (95% CI, 64.2–76.4) in the pembrolizumab group and 62% (95% CI, 0.18–2.60) in the placebo group. Among responders, median DOR was 4.2 months (range, 1.0–26.0) with pembrolizumab and 3.7 months (range, 1.4–25.8) with the placebo. Pembrolizumab plus etoposide and platinum significantly prolonged PFS with a median PFS of 4.5 months versus 4.3 months (HR, 0.75; $p = 0.0023$). The 12-month PFS favored pembrolizumab at 13.6% compared to only 3.1% with the placebo. With a clear PFS benefit, the first co-primary endpoint was met [18].

The second co-primary endpoint was OS. Median OS with pembrolizumab was 10.8 months (95% CI, 9.2–12.9) and with the placebo was 9.7 months (95% CI, 8.6–10.7 months). The difference in OS (HR, 0.80; 95% CI, 0.64–0.98; $p = 0.0164$) did not cross the predetermined threshold for statistical significance. The 12-month OS rate was 45.1% with pembrolizumab and 39.6% with the placebo; the 24-month OS rate was 22.5% with pembrolizumab and 11.2% with the placebo. A consistent effect was observed across the subgroups including platinum choice and PD-L1 expression with the notable exception of patients with baseline brain metastases [18].

There were no new safety signals in KEYNOTE-604. Grade 3 to 4 AEs occurred in 77% in the pembrolizumab group and 75% in the placebo group. Discontinuation of any study treatment due to AE occurred in 14.8% of participants in the pembrolizumab group versus 4.8% in the placebo group. There were no grade 4 or 5 immune-mediated AEs in the pembrolizumab group. The most common immune-mediated AEs were hypothyroidism (10.3% in the pembrolizumab group and 2.2% in the placebo group), hyperthyroidism (6.7% and 2.7%, respectively), and pneumonitis (4.0% and 2.2%, respectively) [18].

**Table 3.** Comparison between IMpower 133, CASPIAN, and KEYNOTE-604 studies.

| Patient and Disease Characteristics at Baseline | IMpower 133 [23] | CASPIAN [17] | KEYNOTE-604 [18] |
|---|---|---|---|
| Therapeutic regimen | Atezolizumab (anti-PD-L1) + carboplatin + etoposide | Durvalumab (anti-PD-L1) + platinum (carboplatin/cisplatin) + etoposide | Pembrolizumab (anti-PD-1) + platinum (carboplatin/cisplatin) + etoposide |
| Patients in the arm of interest, *n* | 201 | 268 | 228 |
| Primary endpoint | PFS, OS | OS | PFS, OS |
| Age groups, *n* (%) | | | |
| <65 years | 111 (55.2) | 167 (62) | 115 (50.4) |
| ≥65 years | 90 (44.8) | 101 (38) | 113 (49.6) |
| Sex, *n* (%) | | | |
| Men | 129 (64.2) | 190 (71) | 152 (66.7) |
| Women | 72 (35.8) | 78 (29) | 76 (33.3) |
| ECOG, *n* (%) | | | |
| 0 | 73 (36.3) | 99 (37%) * | 60 (26.3) |
| 1 | 128 (63.7) | 169 (63%) * | 168 (73.7) |
| Smoking history, *n* (%) | | | |
| Never smoked | 9 (4.5) | 22 (8) | 8 (3.5) |
| Former smoker | 118 (58.7) | 126 (63) | 72 (31.6) |
| Current smoker | 74 (36.8) | 120 (45) | 148 (64.9) |
| Brain or CNS metastasis, *n* (%) | | | |
| Yes | 17 (8.5) | 28 (10) | 33 (14.5) |
| No | 184 (91.5) | 240 (90) | 195 (85.5) |
| PD-L1 status, *n* (%) | | | |
| <1 | 28 (43.8) ** | - | 97 (42.5) *** |
| ≥1 | 36 (56.3) ** | - | 88 (38.6) *** |
| Unknown | - | - | 43 (18.9) *** |
| Duration of follow-up, median | 22.9 mo | 14.2 mo | 21.6 mo |
| ORR | 60.2% (95% CI, 53.1–67.0) | 68% **** Odds ratio 1.56 (95% CI, 1.10–2.22) | 70.6% (95% CI, 64.2–76.4) |
| DOR | 4.2 mo (95%CI, 4.1–4.5) | 5.1 mo (3.4–10.4) | 4.2 mo (1.0+ to 26.0+) |
| PFS, median | 5.2 mo (95% CI, 4.4–5.6) HR 0.77 (95% CI, 0.63–0.95) | 5.1 mo (95% CI, 0.65–0.94) HR 0.78 (95% CI, 0.65–0.94) | 4.5 mo (4.3 to 5.4) HR 0.75 (95% CI, 0.61–0.91; *p* = 0.0023) |
| OS | 12.3 mo (95% CI, 10.8–15.8) HR 0.76 (95% CI, 0.60–0.95; descriptive *p* = 0.0154). | 13.0 mo (95% CI, 11.5–14.8) HR 0.73 (95% CI, 0.59–0.91; *p* = 0.0047) | 10.8 mo (95% CI, 9.2–12.9) HR 0.80 (95% CI, 0.64–0.98; *p* = 0.0164) |
| Any event, *n* (%) | 198 (100) | 260 (98) | 223 (100) |
| Grade 3 or 4, *n* (%) | 134 (67.7) | 163 (62) | 171 (76.7) |
| Immune-related AEs, *n* (%) | 40 (20.2%) | 52 (20%) | 55 (24.7%) |

*n*, number; PFS, progression-free survival; OS, overall survival; ECOG, Eastern Cooperative Oncology Group; CNS, central nervous system; PD-L1, programmed death-ligand 1; ORR, objective response rate; DOR, duration of response; HR, hazard ratio; AEs, adverse events. * World Health Organization (WHO) performance status. ** PD-L1 testing was performed using the PD-L1 immunohistochemical (SP263) assay on a Ventana BenchMark ULTRA automated staining platform according to the manufacturer's instructions. *** PD-L1 status using the combined positive score (CPS), defined as the number of PD-L1-staining cells (tumor cells, lymphocytes, and macrophages) divided by the total number of viable tumor cells, multiplied by 100. **** Objective response by investigator review per Response Evaluation Criteria in Solid Tumors, version 1.1, is defined as patients with complete response or partial response on at least one visit (unconfirmed responses); for confirmed responses, a confirmatory scan was required no sooner than 4 weeks after the initial response.

Pembrolizumab is an active agent in previously treated SCLC, demonstrated by the durable responses seen in the third-line setting. It is difficult to explain the lack of OS benefit in KEYNOTE-604 which stands in sharp contrast to the results of IMpower 133 (atezolizumab) and CASPIAN (durvalumab). There are several important differences between these studies. Both positive trials feature a PD-L1 inhibitor, unlike pembrolizumab, which is a PD-1 inhibitor. This is unlikely to explain these findings, particularly in light of the results from EA5161, a randomized phase II trial that showed the addition of the PD-1 inhibitor nivolumab to platinum plus etoposide improved both PFS and OS [24]. It is possible that the 14-day washout requirement after treatment of brain metastases contributed to worse outcomes in this subset, though the numbers of patients in that group was small. It is also important to note that the rate of effective crossover to immunotherapy can impact OS. In IMpower 133 [23], 15 patients (7.4%) in the control arm received subsequent immunotherapy and the rate was very similar in CASPIAN at 14 patients (5%) [17]. The use of immunotherapy in the control arm was highest in KEYNOTE-604—31 patients (13.9%) in the placebo arm received subsequent immunotherapy [18].

Overall, the difference between a positive and a negative trial can be quite small. The addition of pembrolizumab to chemotherapy improved PFS, and its impact on OS, while not meeting statistical significance, had similar trends to IMpower 133 and CASPIAN. In fact, one patient in KEYNOTE-604 assigned to receive pembrolizumab instead received the placebo. If the analyses were done in the "as-treated" population, the difference in survival would have met statistical significance. While pembrolizumab is unlikely to garner approval in the first-line setting based on this trial, it may serve as an appropriate control arm for future trials.

## 4. Other Therapeutic Treatment Strategies for ES-SCLC

Pembrolizumab has also been studied in the maintenance setting for patients with ES-SCLC after completion of platinum and etoposide-based therapy (four to six cycles) in a phase 2 single-arm trial [21]. Patients that had not progressed after initial chemotherapy were eligible, and there was no entry requirement of PD-L1 expression. Maintenance pembrolizumab 200 mg every 3 weeks began within eight weeks of the last cycle of chemotherapy and continued for 24 months or until disease progression or intolerability. A total of 45 patients were treated with at least one dose of pembrolizumab. Median duration of follow-up was 14.6 months. The median time from the last cycle of chemotherapy to the first dose of pembrolizumab was 5 weeks (range, 3–9 weeks). The median number of cycles administered was 4 (range, 1–26). The ORR was 11.1% (one complete response and four partial responses). Median DOR was 10.8 months (95% CI, 5.8–not reached) [21].

The study failed to meet its primary endpoint with a median PFS of 1.4 months (95% CI, 1.3–2.8), which was similar to historical controls. However, the respective landmark PFS and OS rates of 13% and 37% at one year after initiation of pembrolizumab suggest that the drug did benefit a subset of patients. Only three patients had PD-L1 expression on tumor cells. The PFS duration in these three patients was 10, 11, and 13 months, and two of these three patients were continuing to receive therapy without progression at the time of data cutoff [21]. The authors also suggested benefit in patients with any PD-L1 expression on stromal cells; this was assessed in 20 tumors, of which eight were positive. In these patients, the median PFS was 6.5 months (95% CI, 1.1–12.8) compared to 1.3 months (95% CI, 0.6–2.5) in all patients participating in the study. Differences were also observed in OS with 12.8 months (95% CI, 1.1–17.6) for the patients with PD-L1 expression at the stromal interface versus 7.6 months (95% CI, 2.0–12.7) overall. The safety profile of pembrolizumab as maintenance was similar to previous studies. However, serious AEs included two patients with acute coronary syndrome. In conclusion, maintenance pembrolizumab did not improve median PFS, yet the PFS and OS at 1 year suggest potential benefit in a subset of patients [21].

Paclitaxel in combination with pembrolizumab has been studied. A phase 2, multicenter, open-label, single-arm study evaluated the efficacy of pembrolizumab combined

with paclitaxel in etoposide/platinum-refractory ES-SCLC [22]. The patients received paclitaxel at a dose of 175 mg/m$^2$ every 3 weeks for up to six cycles. Paclitaxel was given alone during the first cycle to increase antigen presentation before pembrolizumab treatment. Pembrolizumab 200 mg IV was added from the second cycle and continued as maintenance monotherapy until disease progression or unacceptable toxicity. Tumor cells were considered positive for PD-L1 if the expression was ≥1% of tumor cells using the Dako 22C3 PD-L1 clone. Of the 26 patients enrolled, 23 patients were evaluable for treatment response. The median follow-up was 11.1 months. The confirmed ORR was 23.1% (95% CI, 6.9–39.3), with 19.2% of patients having a partial response. The median PFS and OS were 5.0 months (95% CI, 2.7–6.7) and 9.1 months (95% CI, 6.5–15.0), respectively. The most common AEs were peripheral sensory neuropathy (57.7%), myalgia (34.5%), and anemia (23.1%) [22].

The trial also analyzed biomarkers including PD-L1 expression, next-generation sequencing, and flow cytometric analysis of peripheral blood immune cells [22]. Twenty-two (85%) patients were PD-L1-negative, and PD-L1 positivity was not found to be significantly associated with PFS (3.9 versus 5.0 months, respectively, $p = 0.897$), while favorable survival outcomes were obtained in patients harboring MET copy number gain (PFS; 3.4 versus 10.5 months, respectively, $p = 0.019$). The tumor mutation burden was not correlated with tumor response nor with the survival outcome. Lower natural killer cell activity after two cycles was significantly associated with tumor response ($p = 0.022$). This study concluded that the combined treatment of pembrolizumab with paclitaxel has a moderate anti-tumor activity [22].

Weiss et al. evaluated the role of pembrolizumab in combination with irinotecan in patients with ES-SCLC (NCT02331251) [25]. The PembroPlus trial was a phase 1b, open-label trial that aimed to identify the recommended phase 2 dose of irinotecan in combination with pembrolizumab. Five patients with ES-SCLC and seven patients with other advanced solid tumors were enrolled in the pembrolizumab plus irinotecan arm. The recommended phase 2 dose of irinotecan was 250 mg/m$^2$ with pembrolizumab on day 1 every 21 days. Four patients had a partial response, one had stable disease, and six presented progressive disease. Immune-related AEs were reported in 33.3% of patients. After mandatory premedication with dexamethasone was initiated, the frequency of grade 3 to 4 AEs decreased [25].

Radiation combinations are also being explored. A single-arm phase 1 trial study assessed the safety of combining pembrolizumab with thoracic radiotherapy after induction chemotherapy for patients with ES-SCLC (NCT02402920) [26]. Pembrolizumab was given every three weeks for up to 16 cycles. Radiotherapy was administered as 45 Gy in 15 daily fractions. A total of 38 patients were enrolled; 33 patients received per-protocol treatment. The median follow-up time was 7.3 months. All patients tolerated pembrolizumab with no dose-limiting toxicity in the 35-day window. There were no grade 4 to 5 toxicities; 6% experienced grade 3 AEs. The median PFS and OS were 6.1 months (95% CI, 4.1–8.1) and 8.4 months (95% CI, 6.7–10.1), respectively. Yet, these rates are difficult to interpret due to heterogeneity in eligibility criteria. This trial concluded that concurrent pembrolizumab and thoracic radiotherapy was tolerated well with few high-grade AEs in the short-term [26].

## 5. Predictive Biomarkers for Immune Checkpoint Inhibitors in SCLC

The survival gains made with the addition of immunotherapy to chemotherapy are modest and likely driven by a subset of patients. There is an ongoing unmet need for predictive biomarkers to identify that patient subset. An imbalance in those patients may explain discordant results between clinical trials but there remains no reliable method to identify patients more likely to achieve long-term survival. Expression of PD-L1 in tumor cells has been shown to be an enrichment factor for the efficacy of PD-1 inhibition in NSCLC, but in SCLC, there is no clear correlation between PD-L1 expression and the effect of immunotherapy [27]. Patients with PD-L1-positive tumors had better PFS and OS in both KEYNOTE-028 and KEYNOTE-158, but there are challenges to using PD-L1

expression as a biomarker in this setting [19,20]. There is a significantly lower prevalence of PD-L1 expression on SCLC tumor cells compared to NSCLC [19,28,29]. Furthermore, PD-L1 expression can be heterogeneous on SCLC tumors [30]. Conflicting data exist regarding the utility of PD-L1 expression for patient selection.

In KEYNOTE-604, analyses of OS by PD-L1 subgroup appeared to show a similar benefit in OS with pembrolizumab plus etoposide and platinum compared with the placebo arm across PD-L1 subgroups [18]. Results from KEYNOTE-604 [18], CASPIAN [17], and IMpower 133 [23] suggest that PD-L1 expression does not appear to be a predictive biomarker for PD-L1/PD-1 checkpoint inhibitor plus chemotherapy in first-line ES-SCLC. An updated exploratory analysis by blood-based tumor mutational burden (bTMB) in IMpower 133 continued to demonstrate that atezolizumab plus carboplatin and etoposide resulted in improved benefit over the placebo arm, independent of bTMB levels [23]. These data suggest that bTMB and PD-L1 status should not be used for patient treatment decisions for the atezolizumab plus carboplatin and etoposide regimen. Although the PD-L1 analyses were limited to a subset of the intention-to-treat population, no numerical difference in efficacy outcomes was observed across the PD-L1 IHC subgroups [23].

Gadgeel et al. explored several factors of the tumor microenvironment as possible predictive biomarkers [21]. They utilized a modified proportion score that incorporated PD-L1 expression in the surrounding stroma. Better PFS and OS were observed in patients with PD-L1 expression surrounding the stroma. However, it is not possible in a single-arm study to determine whether a biomarker has prognostic or predictive utility. The authors suggested that a possible explanation of these differences could be that PD-L1 expression at the stromal interface may represent the presence of effector T-cells in the tumor microenvironment [21].

Exploratory analysis performed in the SCLC cohort of KEYNOTE-158 has shown the potential of the PD-L1 combined score, i.e., the ratio of PD-L1-positive cells, including tumor cells, lymphocytes, and macrophages, to the total number of tumor cells, but further validation is necessary before this is incorporated to the standard-of-care [20].

A recent case-control study investigated the association between immune cell infiltration and SCLC outcome [31]. The study presented a comparative genomic and tumor microenvironment analysis of surgically resected tumors from 23 patients with SCLC who survived at least 4 years after their operation and 18 patients with expected survival ≤2 years. The tumor microenvironment was analyzed by IHC using a panel of immune markers. The IHC was scored in three areas of the tumor—the intraepithelial zone, the tumor stroma zone, and the tumor-no-tumor interface. In all zones, the levels of CD3-, CD8-, and CD4-positive T lymphocytes and lymphocytes expressing PD-1 were significantly higher in long-term survivors than in patients with the expected survival time ($p < 0.01$). The expression levels of both PD-1/PD-L1 in the immune cells were stronger in long-term survivors [31]. Contrarily, the expression of PD-1/PD-L1 in tumor cells was either undetected or very limited, which is consistent with previous reports [31,32]. These data suggest that individualized immunotherapeutic strategies may represent a potentially valuable treatment strategy for SCLC [31]. Furthermore, blockage of alternative immune checkpoints is an area of extensive preclinical investigation. Currently, novel combination approaches are under investigation e.g., inhibition of lymphocyte-activation gene 3 (LAG-3) has shown synergy with PD-1 inhibition in mouse models and enabled more robust T cell responses, suggesting that co-signaling blockade could restore a favorable immune microenvironment that could respond to antigenic stimulation. Therefore, a better definition of the immunogenic microenvironment will clarify the understanding of rational combinatorial strategies. Targeting this foundation could yield the next breakthroughs in the treatment of ES-SCLC [33–35].

Several other biomarkers are currently under investigation, including the presence of positive autoantibodies. Patients with any positive autoimmune antibody (antiSOX2, anti-Hu, anti-Yo, anti-VGCCA, anti-VGPCA, antinuclear, or anti-neutrophil cytoplasmic antibodies) showed a trend for prolonged survival and better PFS in a study combining

carboplatin plus etoposide with ipilimumab [36], but this assessment has not been included in other studies to explore its potential as a biomarker further. Blood-based tests through cell-free tumor DNA profiling are gaining momentum in several cancer types, but unfortunately, no association with response to therapy has been established in patients with SCLC [21].

The presence of a high number of tumor-infiltrating lymphocytes, mismatch repair (MMR) deficiency, or a high frequency of microsatellite instability also predicted improved response to immune checkpoint inhibitors in other types of cancers [27]. KEYNOTE-158, a phase 2 basket trial, demonstrated the clinical benefit of therapy with pembrolizumab among patients with previously treated advanced, high MSI, DNA MMR deficient, non-colorectal cancer; with only four patients with SCLC enrolled in this study, further investigation will be necessary [37].

In summary, there is an ongoing lack of validated and practical biomarkers that can identify the patients with SCLC that will benefit from immune checkpoint inhibitors; prospective studies with control arms are needed to identify these potential biomarkers.

## 6. Future Directions

It is now clear that immunotherapy has a role in the treatment of patients with ES-SCLC. New studies are aiming to potentiate the effects of immune checkpoint inhibitors with the development of new treatment combinations. Select ongoing pembrolizumab trials in patients with SCLC are summarized in Table 4. Currently, pembrolizumab is being studied in combination with radiation, carboplatin, and etoposide in the first-line setting for ES-SCLC (NCT02934503). This study seeks to evaluate pembrolizumab therapy at different times during SCLC treatment—upfront, in conjunction with initiation of chemotherapy, after one cycle of chemotherapy, after completion of first-line chemotherapy (4–6 cycles), or after completion of consolidation thoracic radiation therapy and/or prophylactic cranial irradiation. Pembrolizumab is also being investigated in combination with less conventional regimens such an enzyme replacement therapy (pegzilarginase, NCT03371979), anthracyclines (amrubicin, NCT03253068), vaccines (galinpepimut-S, NCT03761914), T cell amplifier (hyleukin-7, NCT04332653), and bispecific T cell engager (BiTE) (AMG 757, NCT 03319940). Many other compounds and combination regimens are currently under investigation in the field of SCLC. As per a recent review, more than 200 ongoing and recruiting clinical trials are currently evaluating new drugs in SCLC including drugs targeting recurrent genomic alterations, immunotherapeutics, cytotoxics, or antibody–drug conjugates [38–43].

**Table 4.** Select ongoing pembrolizumab trials in small-cell lung cancer.

| Study Phase | Study Name | Clinical Setting | Treatment | Key Endpoints | ClinicalTrials.gov Study Identifier |
|:---:|:---:|:---:|:---:|:---:|:---:|
| 2 | REACTION | 1L concurrent ES-SCLC | Platinum + E +/− pembrolizumab | PFS, OS | NCT02580994 |
| 2 | AFT-17 | 2L platinum-refractory, resistant, sensitive ES-SCLC | Pembrolizumab vs. topotecan | PFS | NCT02963090 |
| 2 | | 1L ES-SCLC | Pembrolizumab + platinum + E + radiation (concurrent, phased, or sequential) | Dynamic PD-L1 expression, PFS and OS | NCT02934503 |
| 2 | | 2L platinum refractory, resistant ES SCLC | Pembrolizumab + amrubicin | ORR | NCT03253068 |
| 1 | MK-3475-011/KEYNOTE-011 | 1L ES-SCLC part E | Pembrolizumab + cisplatin/etoposide vs. pembrolizumab + carboplatin/etoposide vs. pembrolizumab + cisplatin/etoposide + G-CSF | Safety | NCT01840579 |

**Table 4.** *Cont.*

| Study Phase | Study Name | Clinical Setting | Treatment | Key Endpoints | ClinicalTrials.gov Study Identifier |
|---|---|---|---|---|---|
| 1/2 | | 2L platinum refractory, resistant | Pembrolizumab + pegzilarginase | Safety, ORR | NCT03371979 |
| 1/2 | | Refractory to standard therapy | INCAGN01876 + pembrolizumab + epacadostat | Safety, ORR | NCT03277352 |
| 1 | | 3L 2L platinum refractory, resistant, sensitive | Itacitinib + pembrolizumab | Safety | NCT02646748 |
| 1/2 | LUPER | Relapsed after 1L chemotherapy-based regimen | Lurbinectedin + pembrolizumab | Safety, ORR | NCT04358237 |
| 1/2 | | 2L resistant to standard therapy | Galinpepimut-S (vaccine) + pembrolizumab | PFS, OS | NCT03761914 |
| 1b/2a | KEYNOTE A60 | SCLC refractory to checkpoint inhibitor | NT-I7 (hyleukin-7) + pembrolizumab | Safety, ORR | NCT04332653 |
| 1 | | 1L ES SCLC consolidation setting | AMG 757 + pembrolizumab | Safety | NCT03319940 |

1L, first-line; ES-SCLC, extensive-stage small cell lung cancer; E, etoposide; PFS, progression-free survival; OS, overall survival; 2L, second line; PD-L1, programmed death-ligand 1; RP2D, recommended phase 2 dose; ORR, overall response rate; NTC, national clinical trial.

In addition, the incorporation of immune checkpoint inhibitors to the treatment of LS-SCLC is under investigation in several clinical trials, including pembrolizumab (NCT02402920), durvalumab (NCT03811002), and atezolizumab (NCT03540420) [44].

There is a clear need to build upon the early, modest successes of IMpower 133 and CASPIAN with more rational, well-designed studies based on a refined patient selection in order to improve the care of our patients with SCLC.

## 7. Conclusions

In the past three years, immune checkpoint inhibitors have been introduced to the treatment of ES-SCLC, providing our patients with a modest improvement in survival when added to first-line platinum and etoposide therapy. Pembrolizumab was studied in the third-line setting, showing durable responses in a subset of patients and significant improvement of PFS in patients with ES-SCLC, leading to accelerated approval as third-line monotherapy. Pembrolizumab also showed activity in the first-line setting with chemotherapy, improving PFS and narrowly missing statistical significance on an OS benefit. Other pembrolizumab combinations are under active investigation. Despite these advances, SCLC remains a disease with a dismal prognosis, and the role of immune checkpoint inhibitors in further lines of therapy remains unknown. We need to continue efforts to understand the mechanisms underlying both the initial response and the rapid emergence of drug and radiation resistance in SCLC, as this could provide the basis for new treatment strategies.

**Author Contributions:** Literature review, I.R., N.D., S.R.P. and S.V.L.; data analysis, I.R., N.D., S.R.P. and S.V.L.; writing—original draft preparation, I.R., N.D., S.R.P. and S.V.L.; writing—I.R., N.D., S.R.P. and S.V.L. All authors have read and agreed to the published version of the manuscript.

**Funding:** This research received no external funding.

**Institutional Review Board Statement:** Not applicable.

**Informed Consent Statement:** Not applicable.

**Data Availability Statement:** No new data were created in this article. The data supporting this Review are from previously reported studies, which have been cited accordingly.

**Acknowledgments:** The authors thank James P. Zacny, for editorial assistance with manuscript preparation.

**Conflicts of Interest:** N.D. reports consulting and/or advisory board fees from Inivata and As-traZeneca. S.V.L. reports consulting and/or advisory board fees from AstraZeneca, Beigene, Blueprint, Bristol-Myers Squibb, Celgene, G1 Therapeutics, Genentech/Roche, Guardant Health, Inivata, Janssen, Lilly, Merck/MSD, PharmaMar, Pfizer, Regeneron, and Takeda/Ariad and research funding (to institution) from Alkermes, AstraZeneca, Bayer, Blueprint, Bristol-Myers Squibb, Corvus, Debio-pharm, Genentech, Lilly, Lycera, Merck, Pfizer, Rain Therapeutics, RAPT, Spectrum, and Turning Point Therapeutics. The other authors have no conflict of interest to declare.

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
