# Peer review of "Evidence to Date: Evaluating Pembrolizumab in the Treatment of Extensive-Stage Small-Cell Lung Cancer"

_clinpract, doi:10.3390/clinpract11030059_

Round 1

Reviewer 1 Report

In this review article, Riano et al. provide a detailed and comprehensive summary of clinical trials exploring the use of pembrolizumab in small cell lung cancer (SCLC). Overall, this article is excellent and summarizes most of the key references pertaining to pembrolizumab in extensive stage SCLC, and my suggestions are extremely minor and I recommend they be left to the authors' discretion.

I would be interested in hearing more regarding the authors' perspective on the reasons for the less than stellar results of many trials listed, as well as novel combination approaches/strategies to overcome the immune suppressive tumor microenvironment and potentiate the effects of  pembrolizumab-based immunotherapy. 

Author Response

June 9, 2021

Dear editor,

Thank you for forwarding the referee’s comments and the opportunity to revise and resubmit our paper. We have addressed each of these remarks in the point-by-point rebuttal below.

Reviewer #1 (Remarks to the author)

Comment R1. In this review article, Riano et al. provide a detailed and comprehensive summary of clinical trials exploring the use of pembrolizumab in small cell lung cancer (SCLC). Overall, this article is excellent and summarizes most of the key references pertaining to pembrolizumab in extensive stage SCLC, and my suggestions are extremely minor and I recommend they be left to the authors' discretion.

Response R1. Thank you for your comments.

Commentary R2. I would be interested in hearing more regarding the authors' perspective on the reasons for the less than stellar results of many trials listed, as well as novel combination approaches/strategies to overcome the immune suppressive tumor microenvironment and potentiate the effects of  pembrolizumab-based immunotherapy.

Response R2. We have mentioned 2 papers (Gadgeel et al. & Muppa et al.) where tumor microenvironment was studied as possible predictive marker. Based on the reviewer’s comment we have added the following statement:   

Furthermore, blockage of alternative immune checkpoints is an area of extensive preclinical investigation. Currently, novel combination approaches are under investigation e.g., inhibition of lymphocyte-activation gene 3 (LAG-3) has shown synergy with PD-1 inhibition in mouse models and enabled more robust T cell responses, suggesting that co-signaling blockade could restore a favorable immune microenvironment that can respond to antigenic stimulation. Therefore, a better definition of the immunogenic microenvironment will clarify the understanding of rational combinatorial strategies. Targeting this foundation could yield the next breakthroughs in the treatment of ES-SCLC.

Sincerely,    

Ivy Riano M.D.Internal Medicine Resident – PGY3MetroWest Medical Center, Tufts University School of Medicine115 Lincoln Street Framingham, MA, USA 01702Office Phone: 518-383-1000E-mail: [email protected]  

Narjust Duma, M.D.Assistant Professor of Medicine - Thoracic OncologyUniversity of Wisconsin Carbone Cancer CenterK6/544 5666 Clinical Science Center600 Highland Ave Madison, WI 53792T: (608) 265-3837 – Fax: (608) 265-0614E-mail: [email protected]

Reviewer 2 Report

The manuscript entitled “Evidence to Date: Evaluating Pembrolizumab in the Treatment of Extensive-Stage Small-Cell Lung Cancer” by Riano et al., covers clinical data of Pembrolizumab for Extensive-stage Smalll-Cell Lung Cancer. This manuscript is focused and interesting, but need to be revised majorly as follows:

  1. The manuscript is too simple. Giving very narrowed information. There are some published reviews covers on immunotherapy for lung cancer and they already cover pembrolizumab in terms of clinical data and targeted disease. If authors try to focus on the clinical data of pembrolizumab, please try to add more useful and detailed information regarding pembrolizumab including patient numbers, outcomes, possible reasons.
  2. Please add disease information, for broad readership, especially of lung cancer and their stages. And treating history (in introduction.. and/or discussion)
  3. Authors are recommended to refer to many related references. There are 39 references. For review paper, more related and useful information can be drawn from referring many references.
  4. Pembrolizumab was recently approved by FDA, please update the information.

Author Response

RE: clinpract-1234107

Evidence to Date: Evaluating Pembrolizumab in the Treatment of Extensive-Stage Small-Cell Lung Cancer  

Natalie Sun

Assigned Editor

Clinics and Practice

June 11, 2021

Dear editor,

Thank you for forwarding the referee’s comments and the opportunity to revise and resubmit our paper. We have addressed each of these remarks in the point-by-point rebuttal below.

Reviewer #1 (Remarks to the author)

Comment R1. The manuscript entitled “Evidence to Date: Evaluating Pembrolizumab in the Treatment of Extensive-Stage Small-Cell Lung Cancer” by Riano et al., covers clinical data of Pembrolizumab for Extensive-stage Smalll-Cell Lung Cancer. This manuscript is focused and interesting, but need to be revised majorly as follows:

  1. The manuscript is too simple. Giving very narrowed information. There are some published reviews covers on immunotherapy for lung cancer and they already cover pembrolizumab in terms of clinical data and targeted disease. If authors try to focus on the clinical data of pembrolizumab, please try to add more useful and detailed information regarding pembrolizumab including patient numbers, outcomes, possible reasons.

Response R1. We do appreciate the comments by the reviewer and agree that detail information regarding pembrolizumab including patients’ number and outcomes should be included. Pembrolizumab was approved based on evidence provided by two single arm trials: KEYNOTE-028 & 158. We have reviewed both trials in detail, from line 93 through line 130, we have captured the number of patients, baseline characteristic’s, doses, intervals as well as primary and secondary outcomes with safety information. We also included a pooled analysis of these two trials, with pertinent details as mentioned above -lines 132 to 149-.

Commentary R2. Please add disease information, for broad readership, especially of lung cancer and their stages. And treating history (in introduction.. and/or discussion)

Response R2. We have added the following information in the introduction section: Although the tumor–node–metastasis (TNM) classification is preferred to the staging system of the Veterans Administration Lung Study Group (VALSG), which separates limited-stage disease (tumor confined to one hemi-thorax and one radiation port; no malignant pleural or pericardial effusion) from extensive-stage disease (ES) (disease not meeting criteria for limited stage), the latest staging is still widely used in both designing clinical trials and presenting data from them, as it effectively distinguishes patients treated primarily with chemotherapy (limited-stage disease) from those treated with systemic chemotherapy or chemo-immunotherapy (ES disease). The initial approach to SCLC treatment varies substantially by stage. In non-metastatic SCLC, the therapeutic goals are to achieve durable control of thoracic disease and reduce the risk of metastatic dissemination. Localtreatment options include surgery and radiotherapy. Chemotherapy can both augment the local efficacy of radiation and potentially treat micrometastatic disease. The standard chemotherapy regimen in this setting is cisplatin–etoposide. In patients who respond to initial treatment, prophylactic cranial irradiation (PCI) is also part of the standard management with non-metastatic disease. For ES disease, the first-line chemotherapy fornewly diagnosed metastatic SCLC consisted of a platinum agent (cisplatin or carboplatin) withetoposide. Radiotherapy is traditionally reserved for the palliation of symptoms in patients with ESdisease, and PCI remains controversial. 

Commentary R3. Authors are recommended to refer to many related references. There are 39 references. For review paper, more related and useful information can be drawn from referring many references.

Response R3. The number of references depends on how much literature exists on the topic. Since this is a new therapy developed in the latest years, we cited the pertinent sources. We do appreciate this comment, hence we added more references to the manuscript. 

Commentary R4. Pembrolizumab was recently approved by FDA, please update the information.

Response R4. We have mentioned along the manuscript that FDA approved pembrolizumab in the abstract, line 19 as well as in the body of the manuscript, line 155.

Thank you very much for your valuable comments and consideration of this manuscript.

Sincerely,

Ivy Riano M.D.

Internal Medicine Resident – PGY3MetroWest Medical Center, Tufts University School of Medicine115 Lincoln Street Framingham, MA, USA 01702Office Phone: 518-383-1000E-mail: [email protected]  

Narjust Duma, M.D.

Assistant Professor of Medicine - Thoracic OncologyUniversity of Wisconsin Carbone Cancer CenterK6/544 5666 Clinical Science Center600 Highland Ave Madison, WI 53792T: (608) 265-3837 – Fax: (608) 265-0614E-mail: [email protected] 

Reviewer 3 Report

Dr. Riano and colleagues propose a review of the current clinical status of Pembrolizumab Treatment for Extensive-Stage Small-Cell Lung Cancer. SCLC is an aggressive disease with high proliferative rate, early metastasis and poor prognosis therefore, a better understanding of the biology of disease as well presentation of novel therapeutic approaches are very important. In general, the review is well conducted and the findings are well presented, despite the fact that it is not hugely original compared to other literature reviews that are currently around in this field.

Please find below the list of the most important points to be considered while revising the article: 

  • I suggest the authors to report the clinical number according to clinicaltrials.gov for each study they present since will help the readers to better understand this review.
  • Table 1. The authors have titled this table “Comparison of pembrolizumab clinical trials in extensive-stage small-cell lung cancer.” However no results are presented in this table. I suggest therefore to include the main results of each study such as Overall survival , Progression-free survival and Objective response rate.
  • Table 2. The authors should discuss in more detail in the text the pooled analysis, patients selection criteria, treatment etc.,
  • Table 3. This is a comparison of Pembrolizumab, an antibody targeting PD-1, and Atezolizumab, Durvalumab, both targeting PD- L1 I suggest the authors, the table 3 as well the text to be removed from since It does not add any important scientific information in this review. Maybe the authors may discuss these findings in the discussion section, nevertheless, I would not recommend it.
  • Line 267-247: If the analyses were done in the “as-treated” population, the difference 246 in survival would have met statistical significance. This is a statement by the authors and can not be in this section, therefore I suggest to be removed. The authors may discuss this in details in the discussion section.
  • Lines 251-278; I suggest the authors to add the clinical number of this study as well when they present each result.
  • Line 279, the study 18, is combination of pacitaxel plus Ipilimumab; this is not a study that contain Pembrolizumab as treatment therefore should be removed.
  • Lines 313-315. Study 21, the authors as I mention also in a previous comment should add the clinical number for each study they present their results. This study refers to NCT02402920.
  • Lines 352-359. I could not find any association of this paragraph with the rest of the review. The authors should remove it.

Author Response

RE: clinpract-1234107 

Evidence to Date: Evaluating Pembrolizumab in the Treatment of Extensive-Stage Small-Cell Lung Cancer  

Natalie Sun

Assigned Editor

Clinics and Practice

June 11, 2021

Dear editor,

Thank you for forwarding the referee’s comments and the opportunity to revise and resubmit our paper. We have addressed each of these remarks in the point-by-point rebuttal below.

Reviewer #3 (Remarks to the author)

Comment R1. Dr. Riano and colleagues propose a review of the current clinical status of Pembrolizumab Treatment for Extensive-Stage Small-Cell Lung Cancer. SCLC is an aggressive disease with high proliferative rate, early metastasis and poor prognosis therefore, a better understanding of the biology of disease as well presentation of novel therapeutic approaches are very important. In general, the review is well conducted and the findings are well presented, despite the fact that it is not hugely original compared to other literature reviews that are currently around in this field.

Response R1. Thank you for your comments.

Commentary R2. Please find below the list of the most important points to be considered while revising the article: 

I suggest the authors to report the clinical number according to clinicaltrials.gov for each study they present since will help the readers to better understand this review.

Response R2. We have included NCT identifiers for KEYNOTE-028, KEYNOTE-158, IMpower 133, CASPIAN, KEYNOTE-604, pembrolizumab plus irinotecan trial, pembrolizumab plus radiotherapy trial. We previously included identifiers of each clinical trial in table 4, in the right column. Based on your comment, we specifically added the prefix “NCT” followed by the number in table 4. Also, in the section called “Future Directions” from lines 443 to 461, we mentioned the identifiers for each clinical trial. We did not find NCT identifier for 2 trials; one of them was conducted in Korea. Thank you for this observation.

Commentary R3. Table 1. The authors have titled this table “Comparison of pembrolizumab clinical trials in extensive-stage small-cell lung cancer.” However no results are presented in this table. I suggest therefore to include the main results of each study such as Overall survival , Progression-free survival and Objective response rate.

Response R3. The intention of table 1 is to compare the design of published clinical trials that included pembrolizumab. We included the comparison of the main results of key trials such as overall survival, progression-free survival, and objective response rate in table 2. To avoid this confusion, we have renamed table 1 as follows: Table 1. Comparison between designs of published pembrolizumab clinical trials in extensive-stage small-cell lung cancer. 

Commentary R4. Table 2. The authors should discuss in more detail in the text the pooled analysis, patients selection criteria, treatment etc.,

Response R4. Thank you for your comment. We have commented in detail the pooled analysis at the end of the section “Pembrolizumab for previously-treated ES-SCLC”, specifically from lines 153 to 178. We explained that this analysis included 83/131 patients from both cohorts, who had previously received ≥ 2 lines of therapy for advanced disease. We also had discussed main results such as ORR, median DOR, median PFS, and median OS. And, we included details about safety.

Commentary R5. Table 3. This is a comparison of Pembrolizumab, an antibody targeting PD-1, and Atezolizumab, Durvalumab, both targeting PD- L1 I suggest the authors, the table 3 as well the text to be removed from since It does not add any important scientific information in this review. Maybe the authors may discuss these findings in the discussion section, nevertheless, I would not recommend it.

Response R5. Thank you for your comment. Currently, the only immune checkpoint inhibitors approved in the first-line setting of ES-SCLC are durvalumab and atezolizumab. Although these target PD-L1 and pembrolizumab targets PD-1, we find interesting to compare the study designs as the KEYNOTE-604 design (pembrolizumab) was fairly similar to IMpower 133 (atezolizumab) & CASPIAN designs (durvalumab), but the results, unfortunately, were not. To remember this difference in the mechanism of actions to the reader, we included in table 3 the following clarification:  “Atezolizumab (anti-PD-L1) + carboplatin + etoposide”… “Durvalumab (anti-PD-L1) + platinum (carboplatin/cisplatin) + etoposide”… “Pembrolizumab (anti-PD1) + platinum (carboplatin/cisplatin) + etoposide”

Commentary R6. Line 267-247: If the analyses were done in the “as-treated” population, the difference 246 in survival would have met statistical significance. This is a statement by the authors and can not be in this section, therefore I suggest to be removed. The authors may discuss this in details in the discussion section.

Response R6. Thank you for your comment. We agree this is a statement by us, however, this paper does not have a discussion section given the nature of the article. We added at the end of this section “Pembrolizumab for previously untreated ES-SCLC” as part of the discussion of this specific topic. We did not cite references indicating that this is our assertion.

Commentary R7. Lines 251-278; I suggest the authors to add the clinical number of this study as well when they present each result.

Response R7. Thank you for your comment. The article mentioned by the reviewer was published in 2018; unfortunately, within the paper, the authors did not publish the clinical trial identifier. We provide here the link for additional information: https://www.ncbi.nlm.nih.gov/pmc/articles/PMC6833950/

Commentary R8. Line 279, the study 18, is combination of pacitaxel plus Ipilimumab; this is not a study that contain Pembrolizumab as treatment therefore should be removed.

Response R8. We have removed the study with reference #18 as suggested by the reviewer. We have changed the first line of the paragraph as follows:  Paclitaxel in combination with pembrolizumab has been studied. A phase 2, multi-center, open-label, single-arm study evaluated the efficacy of pembrolizumab combined with paclitaxel in etoposide/platinum-refractory ES-SCLC [19].

Commentary R9. Lines 313-315. Study 21, the authors as I mention also in a previous comment should add the clinical number for each study they present their results. This study refers to NCT02402920.

Response R9. We have added the clinical trial number as suggested by the reviewer. A single-arm phase 1 trial study assessed the safety of combining pembrolizumab with thoracic radiotherapy after induction chemotherapy for patients with ES-SCLC (NCT02402920) [21].

Commentary R10. Lines 352-359. I could not find any association of this paragraph with the rest of the review. The authors should remove it.

Response R10. We have removed this paragraph as suggested by he reviewer.

Thank you very much for your valuable comments and consideration of this manuscript. 

Sincerely,   

Ivy Riano M.D.

Internal Medicine Resident – PGY3MetroWest Medical Center, Tufts University School of Medicine115 Lincoln Street Framingham, MA, USA 01702Office Phone: 518-383-1000E-mail: [email protected]  

Narjust Duma, M.D.

Assistant Professor of Medicine - Thoracic OncologyUniversity of Wisconsin Carbone Cancer CenterK6/544 5666 Clinical Science Center600 Highland Ave Madison, WI 53792T: (608) 265-3837 – Fax: (608) 265-0614E-mail: [email protected]

Round 2

Reviewer 2 Report

Authors revised the manuscript accordingly.

Reviewer 3 Report

The manuscript has been sufficiently improved.